# Genetic Algorithms for Interior Comparative Optimization of Standard BCS Parameters in Selected Superconductors and High-Temperature Superconductors

Francisco Casesnoves [1,2]

---

[1] International Association of Advanced Materials, Gammalkilsvägen 18, 59053 Ulrika, Sweden; casesnoves.research.emailbox@gmail.com

[2] Uniscience Publishers, Cheyenne, WY 82001, USA

**Abstract:** Inverse least squares numerical optimization, 3D/4D interior optimization, and 3D/4D graphical optimization software and algorithm programming have been presented in a series of previous articles on the applications of the BCS theory of superconductivity and $T_C$ dual/multiobjective optimizations. This study deals with the comparison/validation of the optimization results using several different methods, namely, classical inverse least squares (ILS), genetic algorithms (GA), 3D/4D interior optimization, and 2D/3D/4D graphical optimization techniques. The results comprise Tikhonov regularization algorithms and mathematical methods for all the research subjects. The findings of the mathematical programming for optimizing type I chrome isotope superconductors are validated with the genetic algorithms and compared to previous results of 3D/4D interior optimization. Additional rulings present a hypothesis of the new 'molecular effect' model/algorithm intended to be proven for Hg-cuprate-type high-temperature superconductors. In molecular effect optimization, inverse least squares and inverse least squares polynomial methods are applied with acceptable numerical and 2D graphical optimization solutions. For the BCS isotope effect and molecular effect, linearization logarithmic transformations for model formula software are implemented in specific programs. The solutions show accuracy with low programming residuals and confirm these findings. The results comprise two strands, the modeling for the isotope effect and molecular effect hypotheses and the development of genetic algorithms and inverse least squares-improved programming methods. Electronic physics applications in superconductors and high-temperature superconductors emerged from the rulings. Extrapolated applications for new modeling for the theory of superconductivity emerged from the numerical and image data obtained.

**Keywords:** interior optimization methods (IO); genetic algorithms (GA); graphical optimization; systems of nonlinear equations; Inverse Tikhonov regularization (ITR); objective function (OF); inverse least squares (ILS); electronic superconductors; high-temperature superconductors (HTSC); BCS theory





## 1. Introduction

A superconductor can be defined as any material type whose electrical resistance is approximately null under specific thermodynamic and electromagnetic conditions. The essential thermodynamic conditions needed to reach the superconductivity state are given by a critical temperature $T_C$, beyond which, toward lower temperatures, a superconductivity effect takes place and interactions with magnetic fields constitute an important modifying factor. The $T_C$ magnitude is around absolute zero Kelvin for conventional superconductors and approximately 100 degrees higher for high-temperature ones. Apart from this crucial condition, there are other physical ones. Namely, the maximum critical current, lower critical magnetic field H, and upper critical magnetic field H. Other factors are pressure and resistivity [1–10]. In general, the large variety of models and formulations within the theory

of superconductivity, cause a multiple factor dependence that constrains the material superconductivity transition/effect [1–10]. The superconductor's principal physical-engineering advantage is its zero-energy loss for electrical currents. However, this excellent property for saving energy is not electromagnetically optimal. The benefit of null-conductivity energy loss is reduced by the necessary energy to cool the material to $-273°$. These antagonist constraints have to be optimized to obtain the most efficient total energy savings.

The current research/theoretical advances in superconductivity are profuse and wide-ranging [1–10]. Their mathematical background is extensive with several theoretical models, approximations, and equation variants [1–10]. At the atomic and molecular level, quantum mechanics and chemistry play a significant role in the basis for the theory of superconductivity [1–10].

High-temperature superconductors are those whose $T_C$ is higher than 80 K. They have an unusually complex molecular composition and several varieties. In the classical BCS theory of superconductivity, the isotope effect model equation reads,

$$[M_i]^\alpha T_{Ci} - K \cong 0 ;$$
$$\text{for i} = 1, \ldots \ldots n ; \tag{1}$$

where

K: Constant parameter. Range specific for every element.
$M_i$: Atomic mass (AMU) of any element isotope of (n) isotopes.
$\alpha$: Constant parameter. Range specific for every element.
$T_C$: Critical temperature (K (usually) or C). Range specific for every element isotopes.
i: Corresponding isotope for every element.

In previous publications, the ILS method was used [1–12] and was based on the Tikhonov regularization theory. The principal difficulty of ILS in TR is the possibility of ill-posed matrices. This can be overcome automatically using modern programming systems, for example, singular value decompositions. The ILS with the $L_2$ norm set as the Tikhonov functions commonly reads

$$\text{minimize functional } J(\alpha),$$
$$J_\alpha(u)_{u \in \Re} = \|Au - K\|_2^2 + \alpha J(u); \tag{2}$$

where

$J_\alpha$ (u): Functions with regularization parameter alpha.
R: Real space.
u: Searched parameter solution.
A: Model matrix vector data.
K: Constant parameter matrix. Range specific for every element.
$\alpha$: Constant parameter. Range specific for every element.
$\alpha 1$: Constant parameter. Tikhonov regularization parameter.
$\| \bullet \|_2$: $L_2$ Norm (at algorithm power 2).

These mathematical parameters have been described in [11–14]. A can be considered in this specific optimization as a model operator matrix. The second term multiplied by $\alpha$ is the regularization parameter. That is, J(u) is the regularization functional term usually related to smoothness, sparsity, and other specific characteristics of the $\alpha$ J (u). The norms are set as $L_1$ or $L_2$ for this research. Instead of using R (real numbers) spaces, the Tikhonov function can be set in Hilbert spaces or C (complex numbers) spaces. A matrix usually requires the decomposition of singular values for better calculations. Since Matlab subroutines have incorporated smoothness, it is taken as $\alpha = 0$ for this study. This Tikhonov model function expressed in a simpler way was developed in previous contributions [2–6,11–15] with acceptable results.

The GA method is a stochastic optimization with differences compared to the ILS method. It is based on Darwin's theory [16,17] of natural selection. The species (pa-

rameters for OF minimization) whose genetic code (magnitudes) results in successful survival/adaptation (OF minimum value) in the environment are selected (parameters for following OF refinement). Therefore, at every step, a selective refinement is performed, discarding the genetic codes (OF parameter numbers) that do not fit the constraints. This process continues until the number of generations of and convergences to the constraints are achieved.

The objectives and innovations of this study were twofold, with the additional aims of the optimization of mathematical modeling and software engineering [1–5,18]. The first was to validate/compare the interior optimization method of previous contributions to genetic algorithm numerical and 3D graphical optimizations. The second was to attempt a tentative application of the isotope effect model of BCS theory on molecular HT superconductors with very similar compositions/molecular structures and critical temperatures. In that case, the model was designated as the molecular effect model. The results for both models were accurate and practical. The molecular effect model for the HTSC Hg-cuprates group showed a parabolic shape, and the $T_C$ theoretical predictions based on this model were obtained. The 2D/3D/4D interior and graphical optimizations showed acceptable imaging and numerical results.

In summary, a comparative study of the different optimization methods was conducted for the chrome and selected HTSCs. The findings were numerically and graphically acceptable and accurate. The molecular effect model simulation results showed very low errors/residuals.

## 2. Mathematical and Computational Methods

Mathematical methods were based on the Tikhonov regularization theory with $L_1$ Chevyshev norms and $L_2$ inverse least squares optimizations [1–11,18]. Numerical data for the algorithm implementation is presented below followed by each method's calculations. Computational methods used were the classical inverse least squares (ILS) and genetic algorithms. ILS was set using several techniques and two norms, $L_1$ and $L_2$. Genetic algorithms method was implemented using an $L_1$ norm. Both optimization methods were intended to be used and the exponential model, logarithmic linearized model, and molecular effect model were optimized with the ILS polynomial-type Equations (1)–(8). The software structure and programming flow chart are explained in Diagram 1.

### 2.1. Numerical Data for Chrome and HTSC Hg-Cuprates

Table 1 shows the numerical data set for all the optimization methods and the two models used. The isotope effect model was applied for the chrome, 3D interior, and GA optimizations. The ILS method was used for the molecular effect model in the HTSC Hg-cuprates.

### 2.2. Mathematical Techniques and Inverse Least Squares Algorithms for Optimization Methods

The BCS isotope effect equation that was set for the optimizations based on Literature and previous studies [1–15] reads

$$[M_i]^\alpha T_{C_i} - K = 0; \tag{3}$$

where

K: Constant parameter. Range specific for every element.
$M_i$: Atomic mass (AMU) of any element isotope of (n) isotopes.
$\alpha$: Constant parameter. Range specific for every element.
$T_C$: Critical temperature (K (usually) or C). Range specific for every element.
i: Corresponding isotope for every element.

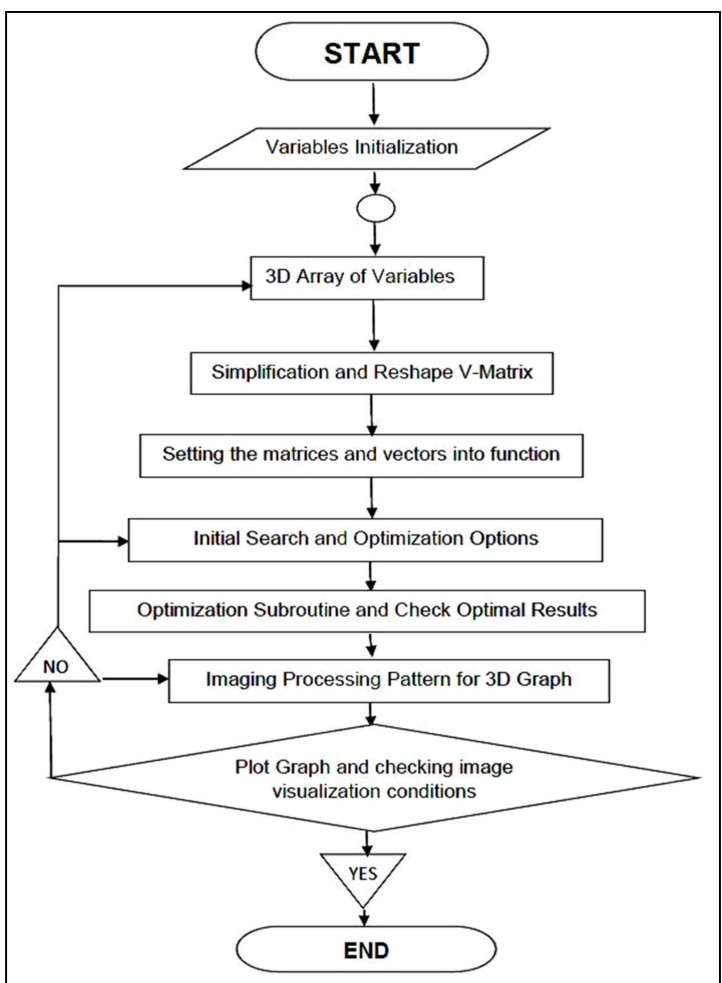

**Diagram 1.** Basic program structure for software. Optimization-specific programs differ for every method/algorithm in Equations (1)–(8). Software, loops, patterns, arrays, and imaging options are improved/developed and adapted on superconductor theory, from a long series of programming subsequent developments [3–11,19–27].

**Table 1.** Numerical data for Cr and HTSC Hg-cuprates.

| NUMERICAL OPTIMIZATION DATA CHROME [SUPERCONDUCTOR, ISOTOPE EFFECT] | | |
|---|---|---|
| **Cr ISOTOPE TYPE BY ATOMIC MASS, (AMU)** | **PERCENTAGE** | **APPROXIMATE $T_C$ (Kelvin)** |
| 52 (NATURAL) | 83.789% | 3 |
| 53 | 9.501% | 3 |
| 54 | 2.365% | 3 |
| 50 | 4.345% | 3 |
| NUMERICAL OPTIMIZATION DATA FOR Hg-CUPRATES [HT-SUPERCONDUCTOR, MOLECULAR EFFECT HYPOTHESIS] | | |
| **FORMULATION** | **MOLECULAR WEIGHT (UAM)** | **APPROXIMATE $T_C$ (Kelvin)** |
| $HgBa_2CuO_4$ | 602.7936 | 97 |
| $HgBa_2CaCu_2O_6$ | 738.42 | 126 |
| $HgBa_2Ca_2Cu_3O_8$ | 874.0432 | 133 |

**Table 1.** *Cont.*

| NUMERICAL OPTIMIZATION DATA FOR Hg-CUPRATES [HT-SUPERCONDUCTOR, MOLECULAR EFFECT HYPOTHESIS] | | |
|---|---|---|
| FORMULATION | MOLECULAR WEIGHT (UAM) | APPROXIMATE $T_C$ (Kelvin) |
| $HgBa_2Ca_3Cu_4O_{10}$ | 1009.7 | 125 |
| $HgBa_2Ca_4Cu_5O_{12}$ | 1145.3 | 110 |
| $HgBa_2Ca_5Cu_6O_{14}$ | 1280.9 | 97 |
| $HgBa_2Ca_6Cu_7O_{16}$ | 1416.54 | 88 |

However, unlike in previous publications [2–4], Equation (1) was also implemented in specific programs with a linear logarithmic transformation such as

$$\ln(T_{Ci}) + \alpha \ln(M_i) - \ln(K) = 0; \tag{4}$$

This linearization showed advantages and disadvantages in the precision of the optimizations. For the molecular effect model, the equation used for the ILS method with an $L_1$ Chevyshev norm reads

$$T_C (MO) \cong \sum_{i=0}^{i=N} a_i [MO]^i; \tag{5}$$

where,

MO: MO is the molecular mass of every compound in the HTSC group selected.
i: N degree of polynomial parameter power. Range [0, N].
$a_i$: Polynomial coefficient. Range [0, N].
$T_C$: Critical temperature (K (usually) or C) for each class of compound.

MO is the molecular mass of the HTSC group selected, in this study, the Hg-Cuprates group. With all these equations, the software algorithms were developed for the program's implementation. The discrete intervals selected for (i) were i $\epsilon$ [0, 5] and i $\epsilon$ [0, 6].

Therefore, based on previous formulas, there were two Tikhonov regularization algorithms applied for the ILS optimization of chrome. The first was the absolute OF value, that is, the Chevyshev $L_1$ norm. The second was with the $L_2$ norm for the Tikhonov function, that is, for the $L_1$ OF algorithm, reads

$$
\begin{aligned}
&\text{minimize Tikhonov functional } J(\alpha), \\
&\text{with} \alpha 1 = 0 \text{and } L_1 \text{ Chebyshev Norm,} \\
&J_\alpha(u)_{u \in \Re} = |Au - K|_{L_1} + [\alpha 1] J(u); \\
&\text{Hence minimize,} \\
&\left| [M_i]^\alpha T_C - K_i \right|_1 \text{ or} \\
&|\ln(T_C) + \alpha \ln(M_i) - \ln(K_i)|_1; \\
&\text{for } i = 1, \dots n \\
&\text{subject to,} \\
&a \leq M_i \leq a_1; \\
&b \leq T_C \leq b_1; \\
&c \leq K_i \leq c_1; \\
&d \leq \alpha \leq d_1;
\end{aligned}
\tag{6}
$$

where

$J_\alpha$ (u): Function with regularization parameter alpha.
R: Real space.
u: Searched parameter solution.
A: Model matrix vector data.
K: Constant parameter matrix. Range specific for every element.
$\alpha$: Constant parameter. Range specific for every element.

$\alpha 1$: Constant parameter. Tikhonov regularization parameter selected null.

$| \bullet |_1$: $L_1$ Chevshev norm (at an algorithm power of 1).

a, $a_1$: Constraint range specified in Table 1.

b, $b_1$: Constraint range specified in Table 1.

c, $c_1$: K optimization parameter range for the program, approximately [20.0, 50.0].

d, $d_1$: $\alpha$ constant range for the program, approximately [0.0001, 0.8].

The constraints a-d are applied for the optimizations [2–4,11,18]. The figure $\alpha 1$ is the Tikhonov regularization parameter. The mathematical concept for the regularization parameter's second term in the TR function means that the minimization of the principal term can reach a better defined/minimized global or local minimum. In other words, the vector/matrix u minimizes the principal term and also the regularization term that depends exclusively on u, which is the solution of the principal term. There is a range of function types for the regularization term [11–13]. This implies that the additional minimization of ($[\alpha 1] \times J(u)$) guarantees/attempts that the most convenient minimum among the minima is determined and is either a global or local minimum. According to the specific conditions for any specific problem, there is a large number of options for the regularization parameter [2–4,11,18].

Similar to Equation (6), the applied algorithm for the ILS with the $L_2$ norm reads

$$
\begin{aligned}
&\text{minimize Tikhonov functional } J(\alpha), \\
&\text{with} \alpha 1 = 0 \text{ and } L_2 \text{ Norm,} \\
&J_\alpha(u)_{u \in \Re} = \|Au - K\|_2^2 + [\alpha 1]J(u); \\
&\text{Hence minimize,} \\
&\left\|[M_i]^\alpha T_C - K_i\right\|_2^2 \text{ or} \\
&\left\|\ln(T_C) + \alpha \ln(M_i) - \ln(K_i)\right\|_2^2; \\
&\text{for } i = 1, \ldots n \\
&\text{subject to,} \\
&a \le M_i \le a_1; \\
&b \le T_C \le b_1; \\
&c \le K_i \le c_1; \\
&d \le \alpha \le d_1;
\end{aligned}
\tag{7}
$$

$J_\alpha(u)$: Function with regularization parameter alpha.

R: Real space.

u: Searched parameter solution.

A: Model matrix vector data.

K: Constant parameter matrix. Range specific for every element.

$\alpha$: Constant parameter. Range specific for every element.

$\alpha 1$: Constant parameter. Tikhonov regularization parameter selected null.

$\| \bullet \|_2$: $L_2$ norm (at an algorithm power of 2).

a, $a_1$: Constraint range specified in Table 1.

b, $b_1$: Constraint range specified in Table 1.

c, $c_1$: K optimization parameter range for the program, approximately [20.0, 50.0].

d, $d_1$: $\alpha$ constant range for the program, approximately [0.0001, 0.8].

### 2.3. Hypothesis and Algorithms for Molecular Effect Model

The isotope effect model is based on the mathematical correlation between the atomic weight of every superconductor element isotope and the critical temperature Tc. This model has proven to be acceptable with some inaccuracies [2–10]. The molecular effect hypothesis proposed here that is mathematically and theoretically presented and numerically simulated is based on a similar modelling criterion, that is, that the HTSCs show several chemical groups whose molecular composition/formulation differ in their proportions of valences/elements [2–10]. From this theoretical basis, it is hypothesized that when deviations in the molecular weight due to proportional/isotopic variations in the molecule

occur, there could be a mathematical model to predict the $T_C$ magnitude change for each HTSC group element.

For this molecular model, the constraint values for the parameters are shown in Tables 1–4. The algorithms set for the ILS molecular effect model with a polynomial $p(MO_i)$ read

$$
\begin{aligned}
&\text{minimize Tikhonov functional } J(\alpha), \\
&\text{with} \alpha 1 = 0 \text{and } L_2 \text{ Norm}, \\
&J_\alpha(u)_{u \in \Re} = \|Au - MO\|_2^2 + [\alpha 1]J(u); \\
&\text{Hence minimize,} \\
&\|T_{Ci} - p(MO_i)\|_2^2 \text{ or} \\
&\|\ln(T_{Ci}) - p(\ln(MO_i))\|_2^2; \\
&\text{for } i = 1, \ldots n \\
&\text{subject to,} \\
&a \leq MO_i \leq a_1; \\
&b \leq T_{Ci} \leq b_1;
\end{aligned}
\tag{8}
$$

$J_\alpha(u)$: Function with regularization parameter alpha.

R: Real space.

u: Searched parameter solution.

$MO_i$: Molecular mass for the HTSC cuprates from Table 1.

$P(MO_i)$: Polynomial optimization parameter matrix of HTSC cuprates range in Table 1.

$\alpha 1$: Constant parameter. Tikhonov regularization parameter, selected null.

$\| \bullet \|_2$: $L_2$ Norm (at an algorithm power of 2).

$a, a_1$: Constraint range specified in Table 1 for the HTSC cuprates.

$b, b_1$: Constraint range specified in Table 1 for the HTSC cuprates.

**Table 2.** Numerical results for Chrome using the GA optimization method.

| NUMERICAL GA OPTIMIZATION RESULTS FOR CHROME FIRST STAGE | | |
|---|---|---|
| Cr ISOTOPE TYPE RANGE (BY ATOMIC MASS, AMU) | K OPTIMAL | OBJECTIVE FUNCTION RESIDUAL ($L_1$ Chebyshev Optimization Norm) |
| [49, 55] | 41.378132 | $176.23 \times 10^{-9}$ |
| NUMERICAL GA OPTIMIZATION RESULTS FOR CHROME SECOND STAGE | | |
| Cr ISOTOPE TYPE RANGE (BY ATOMIC MASS, AMU) | OPTIMAL ALPHA | OBJECTIVE FUNCTION RESIDUAL ($L_1$ Chebyshev Optimization Norm) |
| [49, 55] | 0.6661 | $13.51 \times 10^{-9}$ |

**Table 3.** Numerical results for Chrome using the 3D interior optimization method.

| NUMERICAL 3D/4D INTERIOR OPTIMIZATION RESULTS FOR CHROME FIRST STAGE | | |
|---|---|---|
| Cr ISOTOPE TYPE RANGE (BY ATOMIC MASS, AMU) | K OPTIMAL | OBJECTIVE FUNCTION RESIDUAL ($L_1$ Chebyshev Optimization Norm) |
| [49, 55] | 43.336596 | $7 \times 10^{-3}$ |
| NUMERICAL 3D/4D INTERIOR OPTIMIZATION RESULTS FOR CHROME SECOND STAGE | | |
| Cr ISOTOPE TYPE RANGE (BY ATOMIC MASS, AMU) | OPTIMAL ALPHA | OBJECTIVE FUNCTION RESIDUAL ($L_1$ Chebyshev Optimization Norm) |
| [49, 55] | 0.6794 | $1 \times 10^{-3}$ |

**Table 4.** Numerical results for the ILS molecular effect model for Hg-cuprate HT superconductors.

| NUMERICAL ILS RESULTS FOR MOLECULAR EFFECT MODEL FOR Hg-CUPRATES FIRST STAGE | | |
|---|---|---|
| Hg-CUPRATES MOLECULE TYPE RANGE (BY MOLECULAR MASS, AMU) | OPTIMAL ALPHA | OBJECTIVE FUNCTION RESIDUAL ($L_1$ Chebyshev Optimization Norm, 3000 functions) |
| [738.42, 1416.54] | $5.35 \times 10^{-3}$ | 9.704343 |
| PROGRAMMING FIRST-STAGE DATA | | |
| K | ALPHA | Tc |
| LOG [80, 150] | [0.0001, 1] | [88, 126] |
| NUMERICAL ILS RESULTS FOR MOLECULAR EFFECT MODEL FOR Hg-CUPRATES FIRST STAGE | | |
| Hg-CUPRATES MOLECULE TYPE RANGE (BY MOLECULAR MASS, AMU) | OPTIMAL K | OBJECTIVE FUNCTION RESIDUAL ($L_1$ Chebyshev Optimization Norm, 3000 functions) |
| [738.42, 1416.54] | 109.2585 | 10.45268 |
| PROGRAMMING SECOND-STAGE DATA | | |
| K | FIXED ALPHA | Tc |
| LOG [80, 150] | $5.35 \times 10^{-3}$ | [88, 126] |

MO is the molecular weight of the HTSC selected (i) within an HTSC group and [a, b] are the constraint intervals from Table 1. The other parameters are described in Equations (1)–(7). The constraint values for the parameters are shown in Tables 1–4. All parameter details are described in Equations (1)–(7). OF was chosen either with/without algorithmic linearization, depending on the accuracy of the program results.

In previous publications, [3–6,19], the 2D/3D/4D interior and graphical optimization methods were presented. The authors' definitions of 2D/3D/4D interior optimization were stated:

**Definition 1.** *The interior graphical optimization method, [2–4,22] is a type of nonlinear optimization that combines the separation of variables method with stages of the 3D graphical optimization method.*

For all the algorithms presented in Equations (1)–(8), the 2D/3D/4D interior/graphical optimizations are calculated. The 2D/3D/4D interior optimization method is an improvement of the 3D graphical optimization method [2–4,19–27], set in this article related to superconductors theory [5–10]. Its base is a 2D/3D/4D imaging separation of variables in a series of stages. With the most favorable separation of variables, it is possible to optimize all the parameters throughout the subsequent stages of the 2D/3D/4D optimization plots. In every 2D/3D graph, the most convenient local, global, or semi-local minimum for every OF variable is chosen. The details of this method can be found in [2–4,12,22,26].

*2.4. Genetic Algorithm (GA) Methods*

In brief, the genetic algorithms (GA) optimization method has experienced a recent increase in the use of its optimization variants. Each one of these GA variants has its advantages and disadvantages [10,11]. GA is a stochastic mixed method similar to Monte Carlo but simpler/faster in general.

GA usually selects a randomly large number of successive generations for the objective function minima accuracy subject to constraints. For every generation, three types of choices are applied for the OF. Namely, elite selection, after-mutations, and cross-over changes in the variables' values. The GA method belongs to the stochastic optimization methods group. For instance, it is similar to the random, stochastic simulated annealing (SA) method [10,11,16–24,28]. However, SA cannot determine the global minimum and is stopped at a local minimum function concavity because of its proper algorithm. GA stops when the number of generations constrains and/or the numerical tolerance for a chromosome generation is reached even if that solution is a local or global minimum.

Another random method, Monte Carlo, has a more intensive global minimum search. Monte Carlo estimates the objective function minimum/minima parameters in search of a global minimum. If a global minimum exists, it does not stall easily at any local minima. The Monte Carlo search is exhaustive and this property causes the classical lateness of Monte Carlo. However, new Monte Carlo versions/software have overcome this issue. From a comparison of the genetic algorithm and Monte Carlo methods, the concept emerged that a stochastic optimization method for the development of life and biodiversity was invented and optimized by nature millions of years ago [10,11].

In this study, a simple GA constrained nonlinear optimization was performed. Imaging processing methods were set in some program areas to check/compare the numerical algorithm results. In every GA algorithm step, a numerical refinement was the approach to the OF. There are several variants of the GA method [16,17]. They have in common the steps of selection, mating, mutation, and final convergence. The type applied in this study was the continuous variables GA method, which uses a much larger range of variable numerical data [16,17]. The former decodes the chromosomes and evaluates the OF value for every chromosome in the initial stages. In this study, the continuous GA method was applied.

From Equations (1)–(8), the main difference between the ILS and GA methods can be seen, that is, that the ILS has a matrix with a fixed dataset. In the ILS, matrix A is set to reach the optimal vector matrix solution u for the system A u = K. On the contrary, the GA performs an extensive random search with a set of values that are numerically checked in each step. Those are proven to obtain better OF accuracy subject to constraints remaining throughout the running of the program. Both the GA and ILS methods can be considered useful for optimization; each one shows advantages and disadvantages. Actually, GA methods have been shown to obtain accurate results when the complexity and number of the variables and constraints in the OF increase. A classic example of the Monte Carlo stochastic method is the GEANT systems series, which is generally used in medical physics [25,28,29]. The GEANT4 software applies a large-scale random selection very similar to GA, for instance, to determine the optimal beam radiation parameters in intensity-modulated radiation therapy [28,29]. Both of these methods usually require a longer running time compared to ILS [2–4,13,17,19–27].

### 2.5. GA and Inverse Least Squares Computational Software

Diagram 1 shows the basic structure of the software that was used for programming Equations (1)–(8). The differences between each program are related to subroutines, patterns, loops, matrix definitions, imaging processing subroutines and options, and several others. Genetic algorithm programs [2–4,17] are significantly different but use the same technique. These programs constitute an advance/improvement on the previous studies [2–4,13,17,19–27]. All software for ILS and GA was adapted on superconductors applied theory, materials concepts, Isotope Effect, and optimization fundamentals [30–38].

### 3. Results

The results are divided into two models. The first group is the comparative optimization between the 3D interior optimization and genetic algorithm methods to validate the results of the previous studies for chrome and other superconductors materials [2–4], combined with recent concepts in applied mathematics on modeling and [2–4,39–41]. However, not all the superconductors applied theory are exactly/perfectly coincident along the literature [42–44]. The second group is the inverse least squares method for the hypothesis of the molecular effect model for selected HTSC Hg-cuprates. The numerical results, errors, optimization residuals, and 3D/4D graphics are presented. The running time is specified for each subsection, which is generally consistent with [2–4].

### 3.1. GA Numerical Results for Chrome

Table 2 shows the numerical results using the GA method in two stages as 3D interior optimization was also conducted. The results match well for both methods. The running time was about 4–8 s including the graphics (Intel Core 3).

### 3.2. Interior Tikhonov Optimization Numerical Results for Chrome

Table 3 presents the numerical results for 3D/4D interior optimization for chrome using the isotope effect objective function in logarithmic form. The running time was about 2–4 s including the graphics.

### 3.3. GA Interior Optimization 2D Graphical Results for Chrome

Figures 1–3 show the GA method results for chrome isotopes. The optimization was performed in two stages comprising the best fit, average distance among individuals, and stopping criteria.

### 3.4. Inverse 3D Interior Tikhonov Optimization Graphical Results for Chrome

The image processing method for the 3D interior optimization results for Chrome is shown in Figures 4–7. The numerical data that are presented in the Results section is pictured inset in the 3D charts.

### 3.5. Inverse Least Squares Numerical Results for HTSC Hg-Cuprates with Molecular Effect Model

The calculation using the ILS method for the molecular effect model was conducted with two types of programs. The first was a nonlinear ILS Matlab program based on subroutines. The second was an ILS based on a polynomial fit. The best results were obtained with the second program. Table 4 shows the results for the nonlinear ILS Matlab program. Tables 5 and 6 and Figures 8 and 9 show the results of the ILS polynomial molecular effect model for the HTSC Hg-cuprates. With both methods, the different models were proved, but the polynomial one performed the best. The running time for the ILS method was about 3–6 s, including the graphics, and about 2–5 s for the ILS polynomial method.

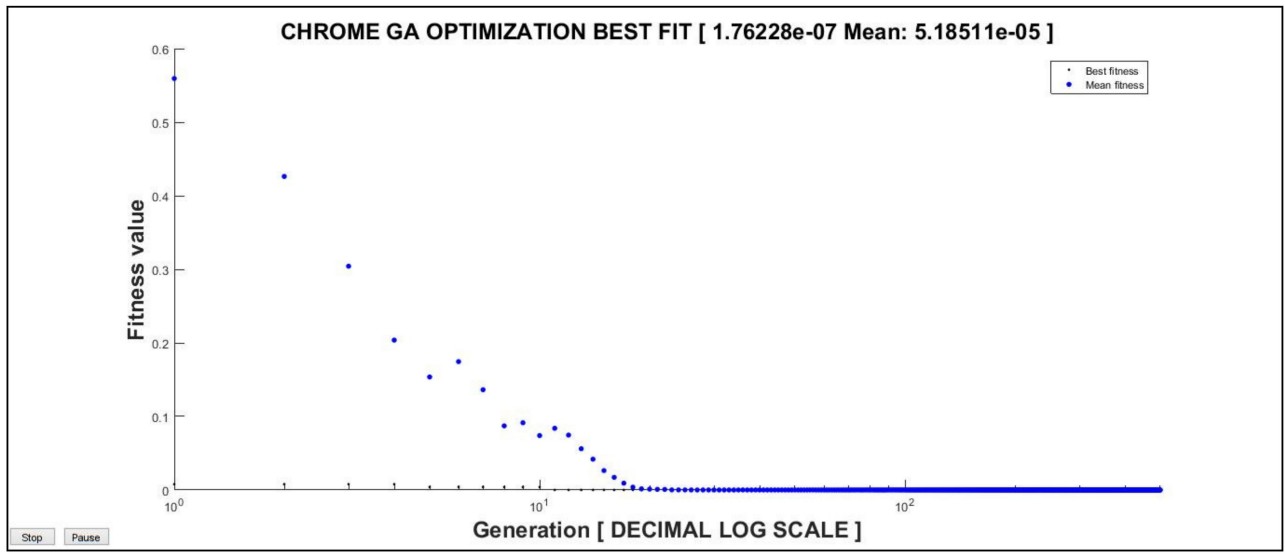

**Figure 1.** Chrome GA results (first stage of optimization) for best fit across generations (500). Results are accurate in magnitude orders. Best Fit results $1.76 \times 10^{-7}$ and Mean $5.19 \times 10^{-5}$.

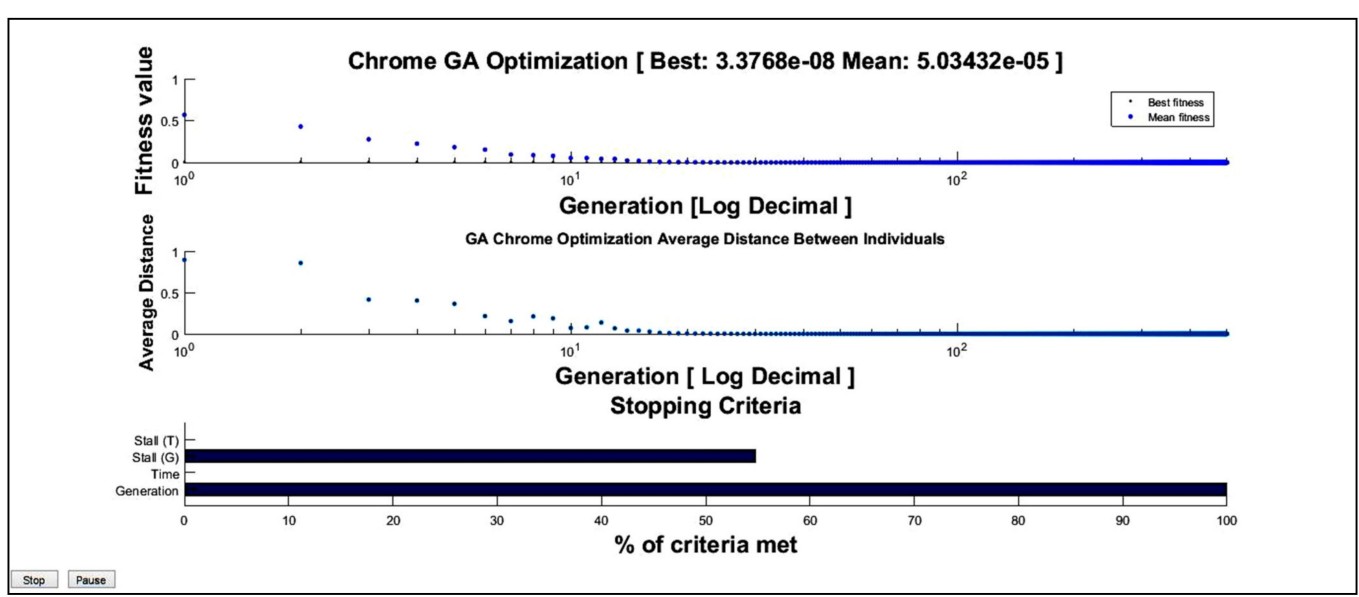

**Figure 2.** Chrome GA (first stage of optimization) combined chart results for best fit across generations (700). Best fit, average distance between individuals, and stopping criteria are shown. Results are accurate. Best Fit and Mean values are lower as it was selected 700 generations.

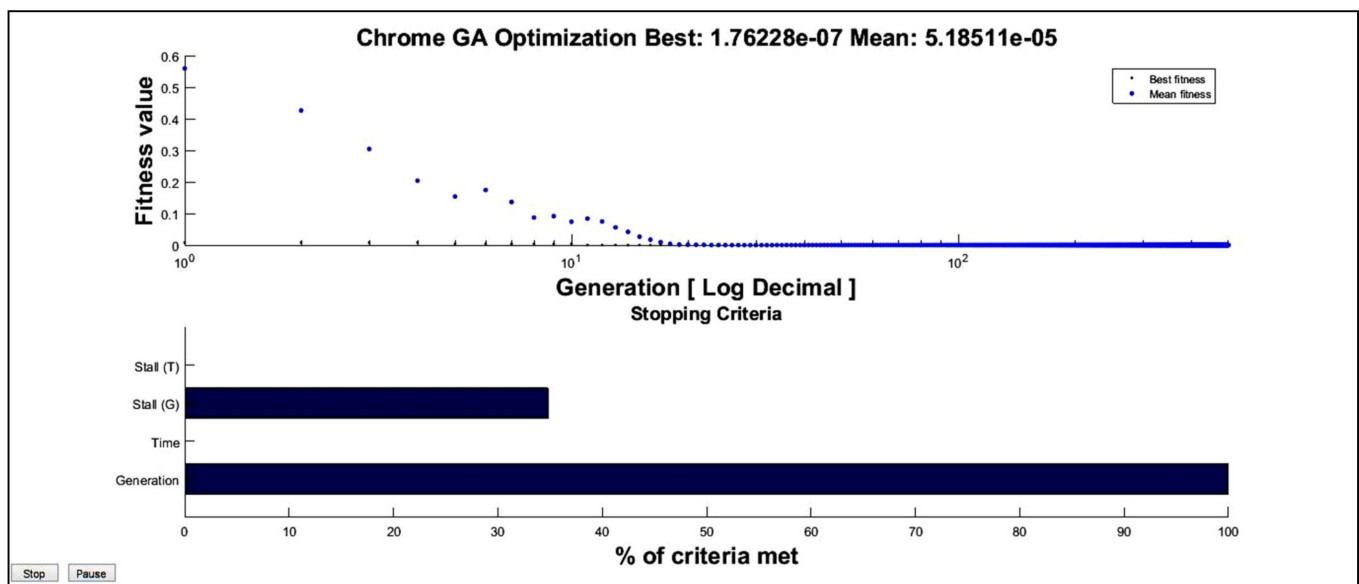

**Figure 3.** Chrome GA (second stage of optimization) combined chart results for best fit across generations (500) and stopping criteria. Results are accurate in magnitude orders. Best Fit results $1.76 \times 10^{-7}$ and Mean $5.19 \times 10^{-5}$.

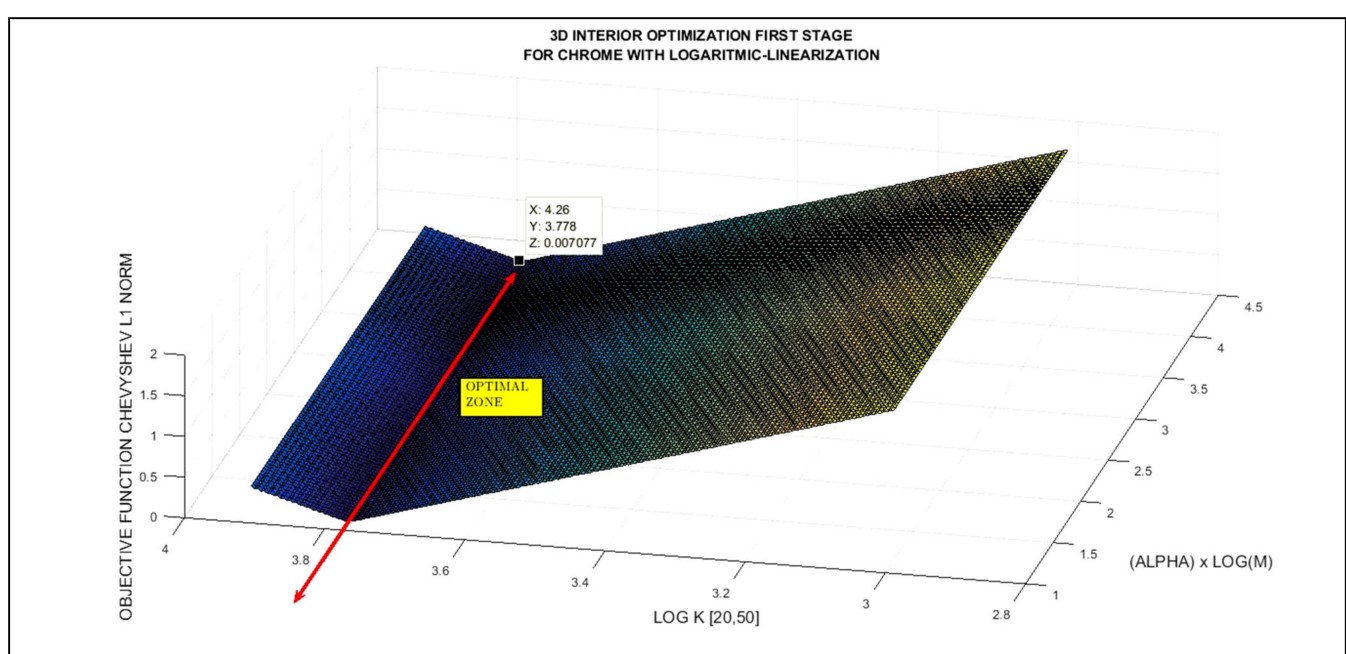

**Figure 4.** Chrome first stage of 3D interior optimization. Optimal zone is accurate and OF magnitude is about $7 \times 10^{-3}$. This value validates the chrome GA results. Optimal zone marked inset. Image processing method 1.

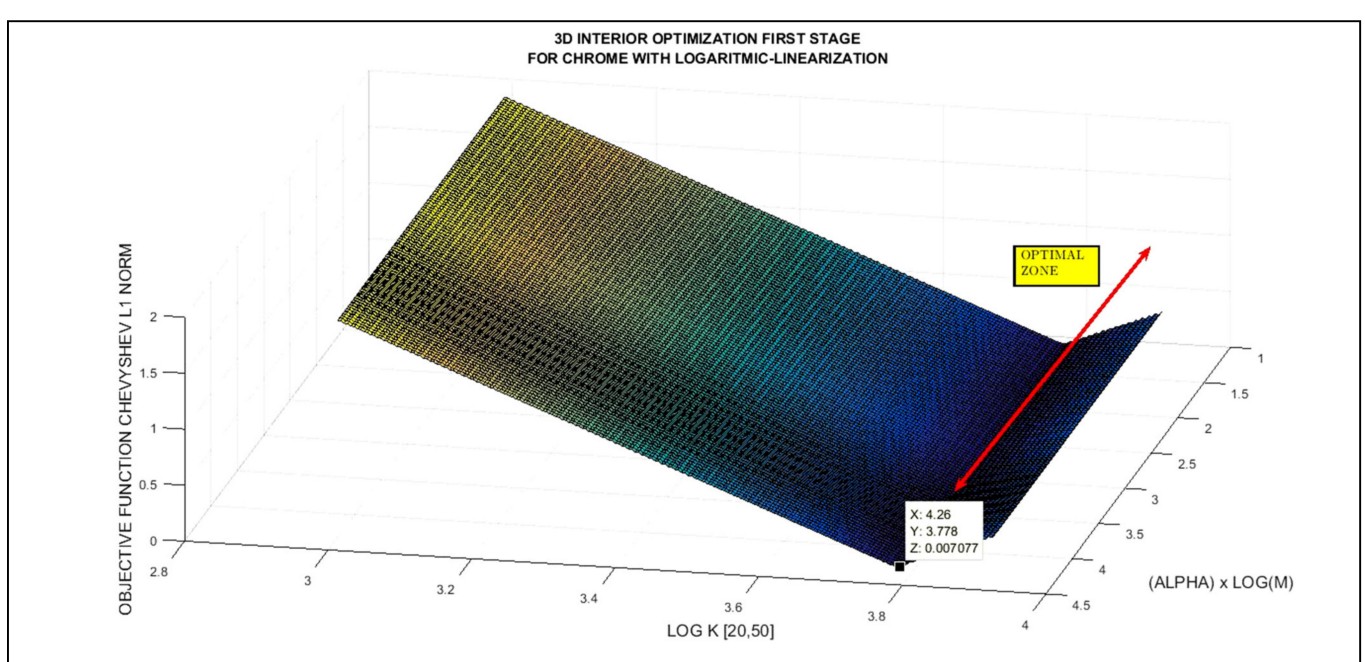

**Figure 5.** Another perspective of chrome first stage of 3D interior optimization. Optimal zone is accurate and residual is about $7 \times 10^{-3}$. This value validates the chrome GA results. Image processing method 1.

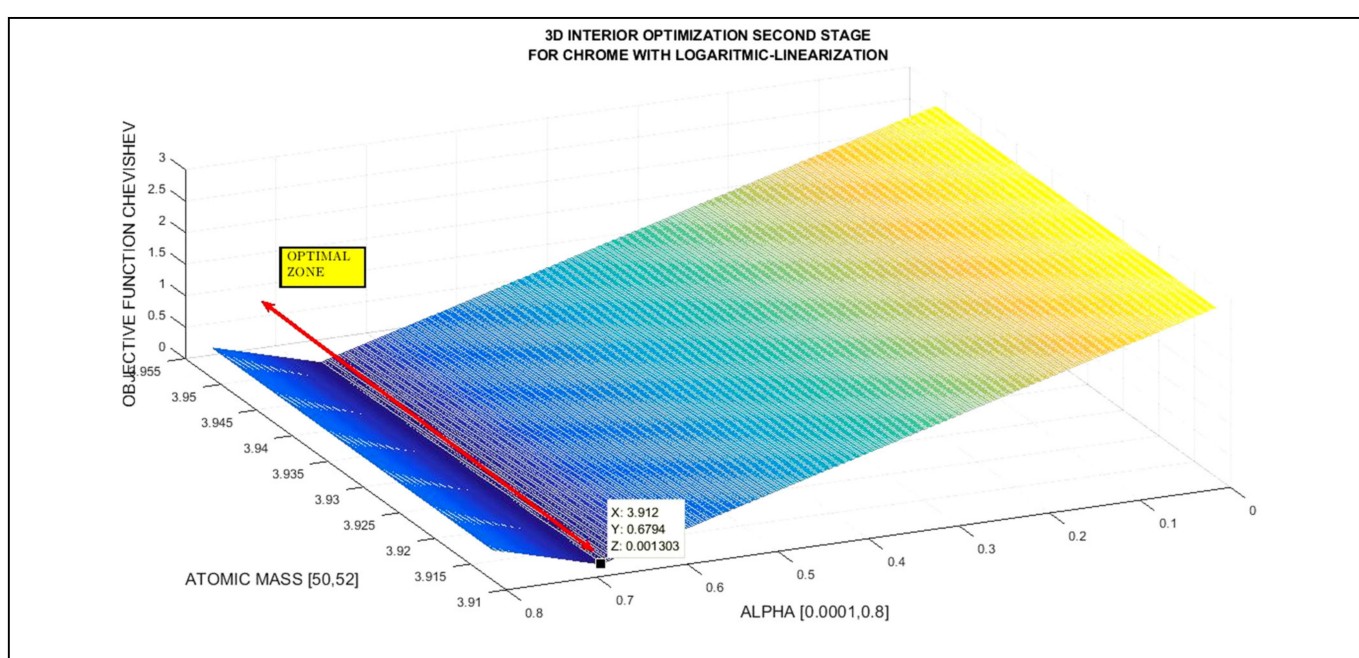

**Figure 6.** Chrome second stage of 3D interior optimization. Optimal zone is accurate and OF magnitude is about $1 \times 10^{-3}$. This value validates the chrome GA results. Subroutine for image processing is different from the first-stage 3D interior optimization charts. Image processing method 2.

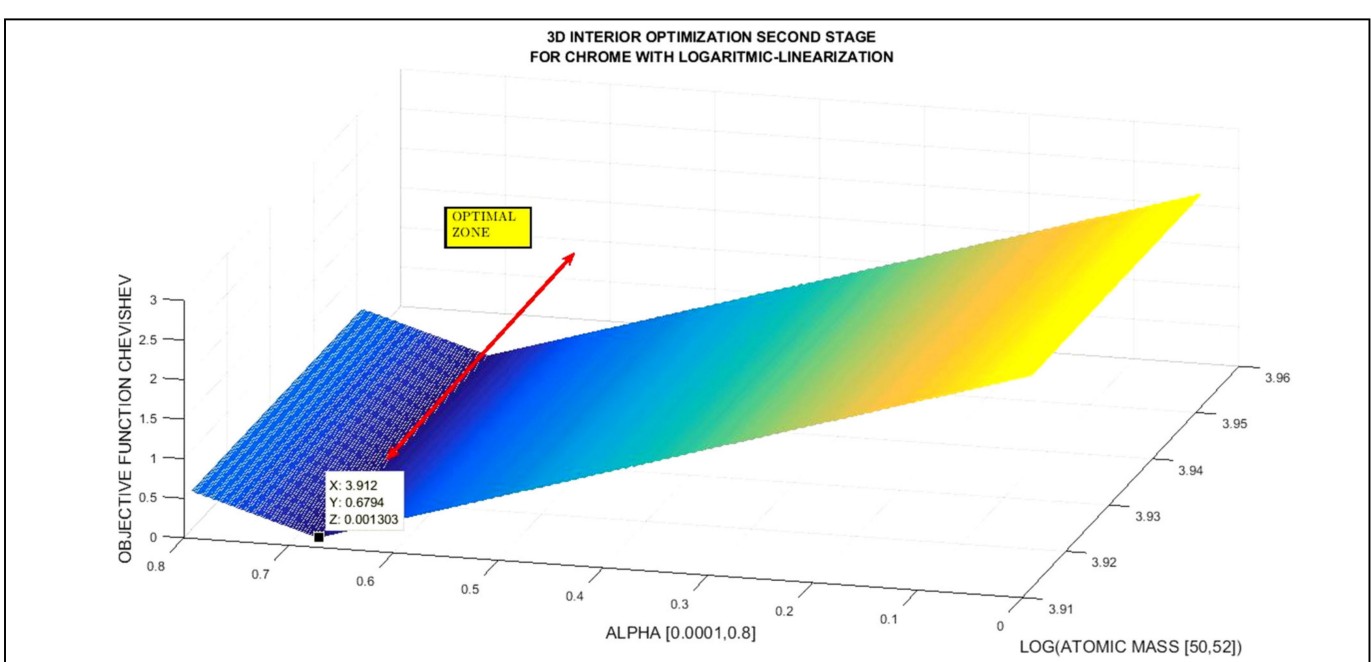

**Figure 7.** Another perspective for chrome second stage of 3D interior optimization. Optimal zone is accurate and OF magnitude is about $1 \times 10^{-3}$. This value validates the chrome GA results. Subroutine for image processing is different from the first-stage 3D interior optimization charts. Image processing method 2.

**Table 5.** Numerical results for the molecular effect model using the 6-degree polynomial method and the approximated equation. For the approximation, a quadratic polynomial is set.

| ILS MOLECULAR EFFECT MODEL 2 (6-DEGREE) | | | |
|---|---|---|---|
| COEFFICIENT | VARIABLE X | COEFFICIENT APPROX | VARIABLE X SELECTED |
| $-1.4683 \times 10^3$ | CONSTANT | $[-1.468]$ | CONSTANT |
| $8.5713$ | X | $[8.571]$ | X |
| $-20.8471 \times 10^{-3}$ | $X^2$ | $[-20.847 \times 10^{-3}]$ | $X^2$ |
| $29.0052 \times 10^{-6}$ | $X^3$ | $[29.005 \times 10^{-6}]$ | $X^3$ |
| $-23.4857 \times 10^{-9}$ | $X^4$ | $0$ | $X^4$ |
| $10.1448 \times 10^{-12}$ | $X^5$ | $0$ | - |
| $-1.7944 \times 10^{-15}$ | $X^6$ | $0$ | - |
| **RESIDUAL = $32.703892 \times 10^{-12}$** | | | |
| **APPROXIMATE POLYNOMIAL** | | | |
| **Tc = $[-1.468] + [8.571]$ MO $+ [-20.847 \times 10^{-3}]$ MO$^2$ $+ [29.005 \times 10^{-6}]$ MO$^3$ $+ [-23 \times 10^{-9}]$ MO$^4$** | | | |

**Table 6.** Numerical results for the molecular effect model using the 5-degree polynomial method and the approximated equation. For the approximation, a cubic polynomial is set.

| ILS MOLECULAR EFFECT MODEL 2 (5-DEGREE) | | | |
|---|---|---|---|
| COEFFICIENT | VARIABLE X | COEFFICIENT APPROX | VARIABLE X SELECTED |
| $4.8106$ | CONSTANT | $[4.811]$ | CONSTANT |
| $-982.4692 \times 10^{-3}$ | X | $[-982.469 \times 10^{-3}]$ | X |
| $4.4871 \times 10^{-3}$ | $X^2$ | $[4.487 \times 10^{-3}]$ | $X^2$ |
| $-6.1759 \times 10^{-6}$ | $X^3$ | $[-6176 \times 10^{-6}]$ | $X^3$ |
| $3.5178 \times 10^{-9}$ | $X^4$ | $0$ | - |
| $-725.5851 \times 10^{-15}$ | $X^5$ | $0$ | - |
| **RESIDUAL = $264.499782 \times 10^{-3}$** | | | |
| **APPROXIMATE POLYNOMIAL** | | | |
| **Tc = $[4.811] + [-982.469 \times 10^{-3}]$ MO $+ [4.487 \times 10^{-3}]$ MO$^2$ $+ [-6176 \times 10^{-6}]$ MO$^3$** | | | |

### 3.6. Numerical Results and Predictive Model Use Verification

The numerical validations/predictions of the optimization methods are presented in Tables 7–9. The validations are also given with the Tc and K predictions. For the molecular effect model, the 6-degree ILS numerical validation is shown in Table 9. Table 7 shows the validations/predictions for the GA chrome results. Table 8 presents the 3D/4D ILS interior optimization method validations/predictions for chrome. Table 9 shows the ILS validations/predictions for the molecular effect model for the HTSC Hg-cuprates.

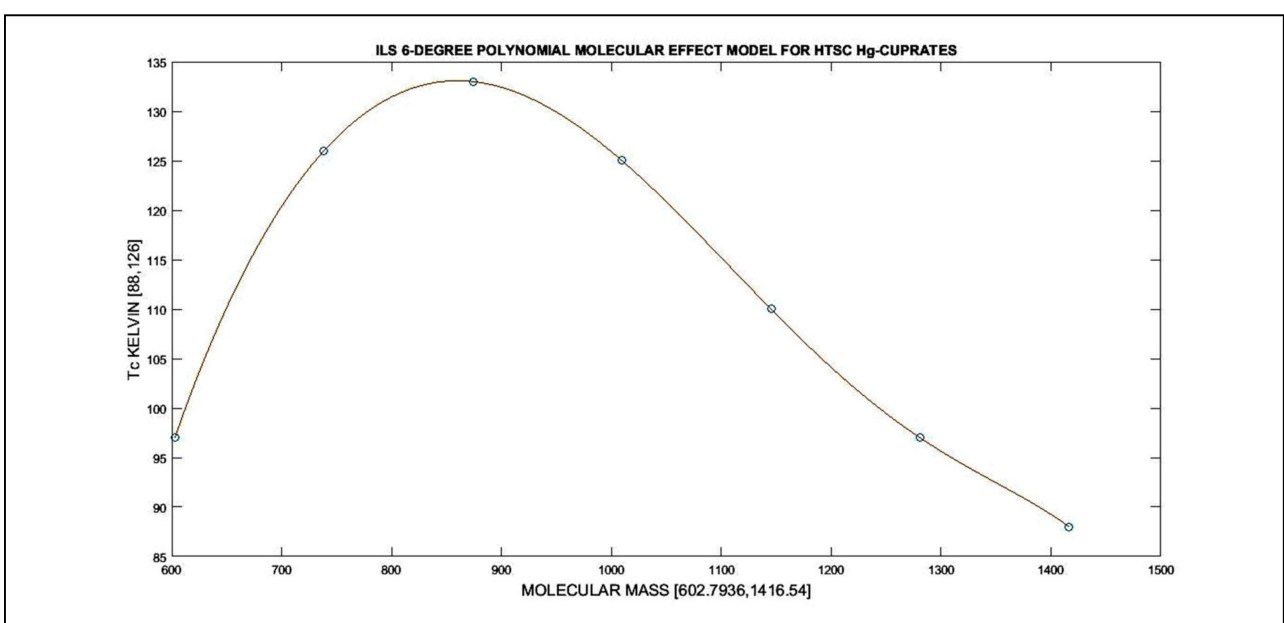

**Figure 8.** Molecular effect model with ILS 6-degree polynomial fit. Generally, Tc matches a parabolic curve when the molecular weight increases.

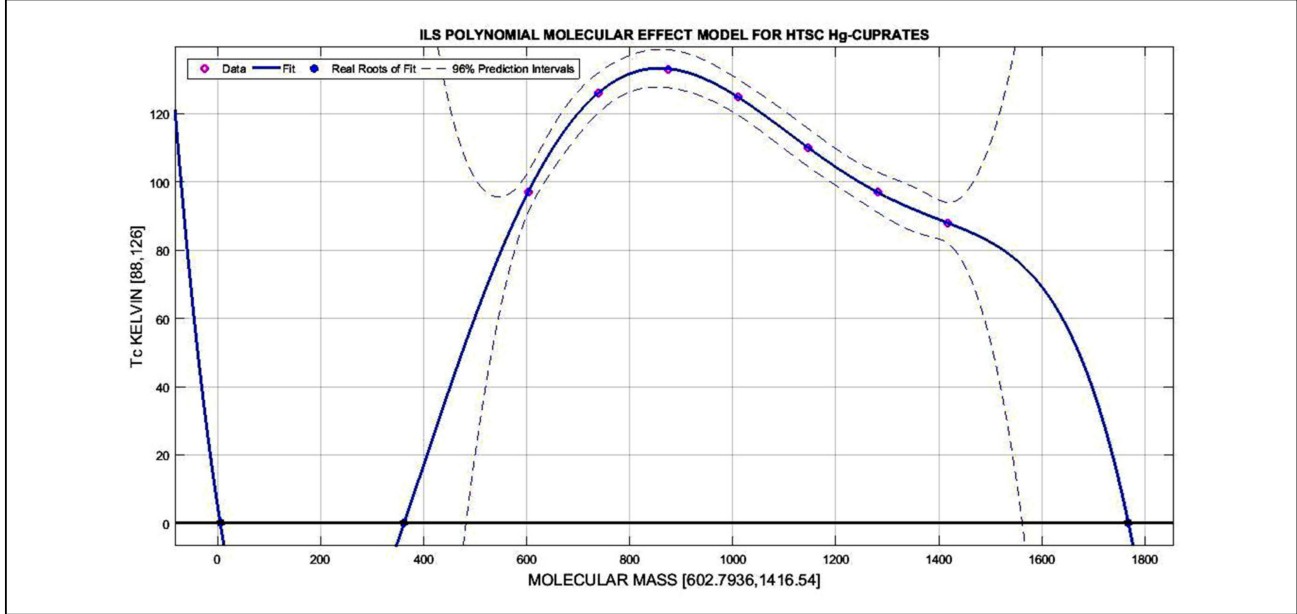

**Figure 9.** Molecular effect model with ILS 5-degree polynomial fit with 96% confidence intervals. Generally, Tc matches a parabolic curve when the molecular weight increases. At lower values of molecular weight, there is a low accuracy of approximately 250 toward lower molecular weight magnitude values.

**Table 7.** GA program numerical results validation for chrome isotope effect model. For UAM the errors decrease.

| CHROME GA NUMERICAL VALIDATION | | |
|---|---|---|
| **ISOTOPE [UAM]** | **K PREDICTED** | **K OPTIMAL BY GA** |
| 50 | 40.6261 | |
| 51 | 41.1655 | 41.3781 |
| 52 | 41.7014 | |
| 53 | 42.2339 | |

**Table 8.** 3D/4D ILS interior optimization numerical validation program numerical results for chrome isotope effect model. For UAM, the errors decrease.

| CHROME 3D/4D ILS INTERIOR OPTIMIZATION NUMERICAL VALIDATION | | |
|---|---|---|
| **ISOTOPE [UAM]** | **K PREDICTED** | **K OPTIMAL BY ILS- INTERIOR OPTIMIZATION** |
| 50 | 42.7958 | |
| 51 | 43.3755 | 43.3365 |
| 52 | 41.9512 | |
| 53 | 42.5240 | |

**Table 9.** Program numerical results for molecular effect model using 6-degree polynomial and approximated equation. Errors are almost null [$10^{-12}$ magnitude order]. Numerical program results for validation simulation give acceptable Tc figures. In this program, MO varied and the corresponding Tc was predicted using the model.

| PROGRAMMING RESULTS FOR ILS MOLECULAR EFFECT MODEL 2 (6-DEGREE) | | | |
|---|---|---|---|
| MOLECULAR WEIGHT (AMU) | Tc EXPERIMENTAL [K] | Tc PROGRAM PREDICTED [MO] | ERROR |
| 602.7936 | 97 | 97.0000 | $9.3223 \times 10^{-12}$ |
| 738.42 | 126 | 126.0000 | $1.8190 \times 10^{-12}$ |
| 874.0432 | 133 | 133.0000 | $3.8654 \times 10^{-12}$ |
| 1009.7 | 125 | 125.0000 | $5.9117 \times 10^{-12}$ |
| 1145.3 | 110 | 110.0000 | $3.6380 \times 10^{-12}$ |
| 1280.9 | 97 | 97.0000 | $6.8212 \times 10^{-13}$ |
| 1416.54 | 88 | 88.0000 | $-2.6375 \times 10^{-11}$ |
| NUMERICAL PROGRAM VALIDATION | | | |
| **MO SIMULATED** | | **Tc PROGRAM PREDICTED** | |
| 602.7936 | | 97.0000 | |
| 750.42 | | 127.3895 | |
| 890.0432 | | 132.7059 | |
| 1029.7 | | 123.0025 | |
| 1180.3 | | 106.1526 | |
| 1295.9 | | 95.9194 | |
| 1480.54 | | 80.5684 | |

## 4. Electronics Physics and Engineering Applications

Applications of 2D/3D/4D interior and graphical optimizations [5–10,31–34,36–38,40–44] are in the field of the BCS theory of superconductivity in the isotope effect model. Further prospective applications for the molecular effect model in the HTSC groups of compounds

are primarily considered for $T_C$ predictions when HTSCs show several chemical groups whose molecular composition/formulation differs in proportion to the valences/elements [5–10,31–34,36–38,40–44].

## 5. Discussion and Conclusions

The objective of this research was to prove/show the similarities in the results of several optimization methods applied for chrome and the HTSC-Hg-cuprates group using the BCS theory of superconductivity [2–4]. For chrome, the methods were the genetic algorithms and 3D/4D interior optimization methods. For the HTSC Hg-cuprates group, a hypothesis for the molecular effect model was approached and numerically analyzed. The rationale for this molecular effect model was set based on the molecules' similar atomic weights (isotope variations in molecular composition and/or molecular approximate proportion/composition for any constituent element) for this HTSC group.

The basis for the molecular effect hypothesis has, therefore, several theoretical applications for Tc and its equation predictions. The first is a prediction of the approximate $T_C$ for a molecule whose composition within the HTSC group differs in the valence/proportion of one/several elements. The second is the case where the molecule is formed by the different isotopes of some/one of its elements, for example, any Hg isotope with a different atomic weight. The third is the case when both the theoretical and experimental facts occur, that is, when both the valence/proportion of one/several elements form part of the molecule and the type of isotopes of the molecule's elements changes. Notably, this study sets a hypothesis/pre-hypothesis based on optimization predictions for the HTSC Hg-cuprates.

The results can be classified into numerical and 3D/2D graphical. The numerical results for the chrome isotope effect, both with GA and 3D/4D interior optimization, can be considered acceptable. Very acceptable numerical and 2D graphical optimization results using the polynomial ILS method from the HTSC Hg-cuprates molecular effect model were obtained with almost zero errors (errors about $[10^{-3}, 10^{-11}]$, Tables 5 and 6, Intel Core-3). The simple ILS method programming for the HTSC Hg-cuprate errors were higher (errors about $[10^{-2}, 10^{-3}]$, Table 4).

In brief, the GA and 3D/4D interior optimization methods have verified previous studies using the 3D/4D interior optimization methods for chrome [2–4]. The GA method is proven as acceptable/accurate (errors about $[10^{-6}, 10^{-7}]$, Table 2, Intel Core-3); the method is in parallel with the 3D/4D interior optimization method. A primary hypothesis for HTSC was tested with the Hg-cuprates group. Both the numerical and graphical results are very acceptable. However, the extension of this molecular effect model to several groups of HTSC remains to be demonstrated.

## 6. Scientific Ethics Standards

The advances in Interior Optimization and Graphical Optimization were created by Dr Francisco Casesnoves on 15 March 2022. The basic 2D/3D graphical optimization methods were created by Dr. Francisco Casesnoves on 3 November 2016, and the interior optimization methods in 2019. The 4D graphical and interior optimization methods were created by Dr. Francisco Casesnoves in 2020. This new GA software was originally developed by the author. This article contains information about previous papers, whose inclusion is essential to make the contribution understandable. The GA nonlinear optimization software was invented/improved based on previous contributions to subroutine modifications, patterns, loops, graphics, and optimal visualizations. In the Introduction section, the paragraph on the basic Tikhonov functional parameters was taken from [11]. The 4D interior optimization method is originally from the author (August 2021). In general, all engineering software constitutes advances/improvements from author's series publications [2–4,12–15,19–27]. This study was carried out and the contents were investigated according to European Union Technology and Science Ethics standards in the *European Textbook on Ethics in Research* from the European Commission, Directorate General for Research, Unit L3, Governance and Ethics, European Research Area, Science and Society, EUR 24452 EN [45–47], which was

based on *The European Code of Conduct for Research Integrity*, revised edition, ALLEA, 2017. This research was conducted entirely by the author, including the computational software used, calculations, images, mathematical propositions and statements, reference citations, and text. When a mathematical statement, proposition, or theorem is presented, a demonstration is always included. If any numerical inconsistency is determined after publication, the corresponding explanations/corrections are included in subsequent articles/books. The article is exclusively scientific, without any commercial, institutional, academic, religious, religious-similar, non-scientific theories, personal opinions, lobbies influences, friens, colleagues or relatives favours, political ideas, or economical influences. When anything is taken from a source, it is adequately recognized. Ideas and some text expressions/sentences from previous publications were emphasized with the aim of clarification [45–47].

**Funding:** This research received no external funding.

**Institutional Review Board Statement:** Not applicable.

**Conflicts of Interest:** The authors declare no conflict of interest.

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
