# Peer review of "Genetic Algorithms for Interior Comparative Optimization of Standard BCS Parameters in Selected Superconductors and High-Temperature Superconductors"

_standards, doi:10.3390/standards2030029_

Round 1
Reviewer 1 Report
Brief Summary
Large variant models and formulations within the theory of superconductors cause multiple-factor constraints for the material superconductivity effect. This paper mainly solves the problem raised in material science regarding the critical temperature as a function of mass and power. Algorithms called ILS and GA are used to solve this problem, and their performances are discussed.
General concept comments
The citations are not appropriately handled, e.g., this [1-9] is cited many times without proper relation to the content, in many cases unnecessary. [3-9][16-23] are works from the same author. This, based on their relevance, are considered self-citing. This is against the publishing rules of MDPI(https://www.mdpi.com/journal/information/instructions).
Many jargons undefined: GA, OF, even BCS. Check whether you use jargons before the definition appears.
Maybe the editors can help with some format issues, especially for the equations: they need to be succinct in forms, define the variable range in the text and always define every parameter that appears.
In the text, there are other references that are name-and-year based, instead of numerical, which is not consistent.
As author claimed the improvement in computing speed, no data was given in this aspect.
Some wording like “Results show be accurate” needs to be improved.
Specific comments
Title too long.
L72: Eqn1, explanation for \alpha missing
L95: “C ones”?
L135: why cite [1-9, 10-12]?
Table 1: UAM->AMU
L293: “[Casesnoves, 2019,..”, is this a quote?
L337-339: explanation of Binary GA is not necessary.
L352: cite GEANT properly
L416->matc->match
Diagram 1: a “YES” is missing, and a condition should be placed in the diamond box.
Author Response
REVIEWER REPORT
IT IS SUBMITTED IN PDF, ALL CORRECTIONS DONE AND NEW HIGH-IMPACT BOOKS ARE ADDED TO REFERENCES, IMPROVED IMAGES ALSO
REVIEWER 1 REPORT CORRECTIONS
[ ATTACHED ALSO THE PAPER WITH ALL THEM IMPLEMENTED ]
REVIEWER 1
Open Review
(x) I would not like to sign my review report
( ) I would like to sign my review report
English language and style
( ) Extensive editing of English language and style required
(x) Moderate English changes required
( ) English language and style are fine/minor spell check required
( ) I don't feel qualified to judge about the English language and style
|
Yes |
Can be improved |
Must be improved |
Not applicable |
|
(x) |
( ) |
( ) |
( ) |
|
( ) |
( ) |
(x) |
( ) |
|
( ) |
( ) |
( ) |
(x) |
|
( ) |
(x) |
( ) |
( ) |
|
( ) |
(x) |
( ) |
( ) |
|
( ) |
( ) |
( ) |
(x) |
Comments and Suggestions for Authors
Brief Summary
Large variant models and formulations within the theory of superconductors cause multiple-factor constraints for the material superconductivity effect. This paper mainly solves the problem raised in material science regarding the critical temperature as a function of mass and power. Algorithms called ILS and GA are used to solve this problem, and their performances are discussed.
General concept comments
1 The citations are not appropriately handled, e.g., this [1-9] is cited many times without proper relation to the content, in many cases unnecessary.
DONE CITATIONS WITH AUTHOR NAME ARE REMOVED, CITATIONS [1-9] ARE SUBSTITUTED BY [3-8 (series of superconductors papers, and sometimes 6 (doctoral thesis software/methods ILS), and [16-24]. [16-24] is a series of consecutive papers that contained evolution of the software, imaging processing, and algorithms methods. NEW high- impact books of prestigious editors are set in refs [43-47]. ORDER OF REFERENCES WILL BE SET IN FOLLOWING REVIEW/EDITION.
2 [3-9] [16-23] are works from the same author. This, based on their relevance, are considered self-citing. This is against the publishing rules of MDPI(https://www.mdpi.com/journal/information/instructions).
RELEVANCE OF AUTHOR CITATIONS
The relevance of author citations are related to two topics
1 Superconductors previous papers before reaching the new method of genetic algorithms and new Tikhonov algorithms.
2 a series of previous papers related to evolution of original software of the author. These citations relevance is related to optimization software and methods made by author.
Apart from that
The logarithmic linearization comes from other publications not in superconductors and is a method that has to be cited.
NOTE
THE ORDER OF REFERENCES WILL BE SET PROPERLY DURING FOLLOWING REVIEW STAGES, NOW WE ARE COMPLETING NEW REFERENCES
NEW REFERENCES ADDED
BOOKS [these books have essential concepts and information along several chapters for introduction that was used for the article first submission, but they were not included in first submission references]
1 Seidel, P. Applied superconductivity. Volume 1 and 2. Wiley-VCH. 2015.
2 Buschow, K. Magnetic & superconducting materials. Second edition. Elsevier. 2003.
3 Drechsler, S ; Mishonov, T. High-T c Superconductors and Related Materials
Material Science, Fundamental Properties, and Some Future Electronic applications. Springer science media, B.V. 1998.
4 Parinov, I. Microstructure and Properties of High-Temperature Superconductors. Second Edition. 2017.
5 Plakida, N. High-Temperature Cuprate Superconductors Experiment, Theory, and Applications. Springer Series in Solid-State Sciences. 2010.
6 Fossheim, K ; Sudbø, A. Superconductivity Physics and Applications. Wiley. 2004.
7 Wang, Y. F. Fundamental elements of applied superconductivity in electrical engineering. Wiley. 2013.
The following references for GEANT [GEANT4] were set in references,
1 Pia, M, and Colls. The Geant4 Simulation Toolkit. IEEE Nuclear Science Symposium and Medical Imaging Conference. Seoul. 2013.
2 Allison, K, and Colls. Geant4 Developments and Applications. IEEE Transactions on Nuclear Science, vol. 53, no. 1. February 2006.Many jargons undefined: GA, OF, even BCS. Check whether you use jargons before the definition appears.
3 jargons are set in keywords and defined the first time when the term appears
DONE
BCS is the widely usual at literature notation for the prize-inventors of that theory, I think it needn’t be clarified, those inventors were John BARDEEN, Leon COOPER Robert SCHRIEFFER ‘BCS’ initials theory
it is set in keywords, if reviewer wants it in other place, please say that,
4 Maybe the editors can help with some format issues, especially for the equations: they need to be succinct in forms, define the variable range in the text and always define every parameter that appears.
THE SETTING OF EQUATIONS PARAMETERS AND VARIABLES ARE CHANGED ACCORDING TO STANDARD PAPERS/BOOKS,RANGE IS SET ALSO
FOR EXAMPLE EQUATION (1)
where
K : Constant parameter. Range specific for every element.
Mi :Atomic Mass (AMU) of any element isotope if (n) isotopes.
α : Constant parameter. Range specific for every element.
TC : Critical temperature (K (usually), or C). Range specific for every element.
i : Corresponding isotope for every element.
EXAMPLE EQUATION (2)
where,
Jα (u) : Functional with regularization parameter alpha.
R : Real space.
u : Searched parameter solution.
A : Model matrix-vector data.
K : Constant parameter matrix. Range specific for every element.
α : Constant parameter. Range specific for every element.
‖ • ‖2 : L2 Norm (at algorithm power 2).
EQUATION 4 LOGARITMIC HAS SAME NO NEED FOR EXPLANATION
FOR EXAMPLE EQUATION (6)
where,
Jα (u) : Functional with regularization parameter alpha.
R : Real space.
u : Searched parameter solution.
A : Model matrix-vector data.
K : Constant parameter matrix. Range specific for every element.
α : Constant parameter. Range specific for every element.
α1 : Constant parameter. Tikhonov Regularization Parameter, selected null.
| • |1 : L1 Chevshev Norm (at algorithm power 1).
a,a1 : Constraint range specified at Table 1.
b, b1 : Constraint range specified at Table 1.
c, c1 : K optimization parameter range for program approximately [20.0,50.0].
d,d1 : α constant range for program approximately [0.0001,0.8 ] .
5 In the text, there are other references that are name-and-year based, instead of numerical, which is not consistent.
DONE
References to my name came from my personal drafts, I had to submit the paper and there was not time to remove them; they are substituted by new important books, if it is the consistent case, that give more reliability to the paper [ new refs, 39-47]. If those refs are not important, are removed. Reference [3-9] is used frequently to mark the ILS method that was used in series of superconductor papers and doctoral thesis [ doctoral thesis is based on ILS optimization about 12 papers]. GEANT references are set [ 46,47 ]. A series of references [16-24] are frequently used because the progressive software methods for those papers are used as experience in programming for this paper. For example, paper [23] was very important to begin to develop ILS method in programming.
THESE NEW REFERENCES< IMPORTANT BOOKS HIGH IMPACT, ARE
39. Buschow, K. Magnetic & superconducting materials. Second edition. Elsevier. 2003.
40. Seidel, P. Applied superconductivity. Volume 1 and 2. Wiley-VCH. 2015.
41. Drechsler, S ; Mishonov, T. High-T c Superconductors and Related Materials Material Science, Fundamental Proper-ties, and Some Future Electronic applications. Springer science media, B.V. 1998.
42. Parinov, I. Microstructure and Properties of High-Temperature Superconductors. Second Edition. Springer. 2017.
43. Plakida, N. High-Temperature Cuprate Superconductors Experiment, Theory, and Applications. Springer Series in Solid-State Sciences. 2010.
44. Fossheim, K ; Sudbø, A. Superconductivity Physics and Applications. Wiley. 2004.
45. Wang, Y. F. Fundamental elements of applied superconductivity in electrical engineering. Wiley. 2013.
46. Pia, M, and Colls. The Geant4 Simulation Toolkit. IEEE Nuclear Science Symposium and Medical Imaging Confer-ence. Seoul. 2013.
47. Allison, K, and Colls. Geant4 Developments and Applications. IEEE Transactions on Nuclear Science, vol. 53, no. 1. February 2006.
6 As author claimed the improvement in computing speed, no data was given in this aspect.
DONE
SETTING IN TEXT SPEED
Section 3.1. Set: Running time is about 4-8 seconds, included graphics.
Section 3.2. Set : Running time is about 2-4 seconds included graphics.
Section 3.5. Set : Running time is about 3-6 seconds, included graphics, for ILS, and ILS-Polynomial is about 2-5 seconds.
Running times are usually,
TIME FOR ISOTOPE EFFECT CHROME (Intel Core 3)
ILS METHOD : about 2-4 seconds included graphics (logarithmic form).
GA METHOD : about 4-8 seconds, included graphics
TIME FOR MOLECULAR EFFECT A BIT LONGER (Intel Core 3)
ILS METHOD : about 3-6 seconds included graphics
POLYNOMIAL METHOD : about 2-5 seconds, included graphics
7 Some wording like “Results show be accurate” needs to be improved.
DONE
At Conclusions the following phrase was changed
‘ GA method is proven as an acceptable/accurate [ errors about [ 10-6 , 10-7 ], Table 2], method in parallel with 3D/4D Interior Optimization. ‘
At Conclusions the following phrase was changed
Very acceptable numerical and 2D Graphical Optimization with polynomial ILS method results for the HTSC Hg-Cuprates Molecular Effect were obtained with almost zero errors [ errors about [ 10-3 , 10-11 ], Tables 5,6,9 Intel Core-3], simple ILS method programming for HTSC Hg-Cuprates errors were higher [ errors about [ 10-2 , 10-3 ], Table 4 ]. Note, in Table 4 the total residual is divided by functions number 3000 to calculate approximate error.
At Table 9 footnote the following phrase was changed
Table 9.- Program Numerical results for Molecular Effect model with 6-degree polynomial and approximated equation. Errors are almost null [10-12 magnitude order].
Specific comments
8 Title too long. DONE
PREVIOUS TITLE
Mathematical Genetic Algorithms to 3D Interior and Inverse Least Squares comparative optimization for standard BCS-parameters Modelling in selected Superconductors and High-Temperature Superconductors
TITLE CORRECTED SHORTENED
Genetic Algorithms to Interior comparative optimization for standard BCS-parameters in selected Superconductors and High-Temperature Superconductors
AFFILIATIONS TEXT REDUCED ALSO
9 L72: Eqn1, explanation for \alpha missing
DONE
It was only alpha, alpha1 set,
10 L95: “C ones”?
There is a misunderstanding, the phrase was
‘Instead R Space, Hilbert Spaces or C ones’
That means that functional of algorithm can be defined in
real numbers spaces
Hilbert spaces (which are complex numbers usually , that is space ‘C’, Hilbert Spaces have strict mathematical conditions added
C ones means Complex number spaces
NEW PHRASE
‘Instead R (real numbers) space, the Tikhonov Functional can be set in Hilbert spaces or C (complex numbers) ones’
11 L135: why cite [1-9, 10-12]?
DONE REMOVED THAT CITATION
Explanation
Yes it was reiteration , what was intended to mean was the programming methods that are developed in successive publications, from [3] to [9] (included my PhD thesis in [6]. But it was not necessary.
The evolution of my software programs is paper after paper, and not in the same topic, so that is the reason for so many self citations (reduced in reviewed paper)
12 Table 1: UAM->AMU
DONE CORRECTED THIS TYPING MISTAKE WAS CORRECTED IN ALL TABLES ALSO
13 L293: “[Casesnoves, 2019,..”, is this a quote?
DONE EXPLAINED TO REVIEWER
The text is before Graphical Optimization Definition. Graphical Optimization was invented originally in my PhD Thesis, other authors speak about Graphical Optimization but those are different methods. The initial text was
In previous publications, [1-9], 2D/3D/4D Interior and Graphical Optimization 292 Methods were presented, [Casesnoves, 2019, 1-9], 2D/3D/4D Interior Optimization [1-9] 293 was defined,
CORRECTION
‘In previous publications, [3-6,19], 2D/3D/4D Interior and Graphical Optimization Methods were presented. Author’s definition of 2D/3D/4D Interior Optimization was stated, ‘
14 L337-339: explanation of Binary GA is not necessary. DONE
Only this sentence for Continuous-GA at text:
The type applied in this study is Continuous-variables GA method that uses much longer range of variable numerical data [10-11].
15 L352: cite GEANT properly
DONE
The following references for GEANT [GEANT4] were set in references [46,47],
1 Pia, M, and Colls. The Geant4 Simulation Toolkit. IEEE Nuclear Science Symposium and Medical Imaging Conference. Seoul. 2013.
2 Allison, K, and Colls. Geant4 Developments and Applications. IEEE Transactions on Nuclear Science, vol. 53, no. 1. February 2006.
The following text about GEANT4-monte Carlo was set in paper,
A classical example of Monte Carlo stochastic method is GEANT systems series, generally used in Medical Physics [46,47]. GEANT4 software applies large-scale random selection very similar to GA, for instance, to determine the optimal beams radiation parameters in Intensity Modulated RadiationTherapy [46,47].
16 L416->matc->match
DONE: NEW TEXT
‘Table 2 shows numerical results for GA method in two stages as 3D Interior Optimization was also done. Results match well for both methods.’
17 Diagram 1: a “YES” is missing, and a condition should be placed in the diamond box.
DONE FLOW CHART CORRECTED, SEE
Submission Date
14 March 2022
ADDITIONAL EXPLANATIONS FOR REVIEWER 1
Reviewer 2 wanted an improvement in GA graphics, that required the study of imaging processing for better pics and scale changes. Those new graphs according to Reviewer 2 are
X AXIS LOG [DECIMAL] SCALE DONE
Figure 1 here CORRECTED
Figure 1.- Chrome GA results (first stage optimization) for best fit along generations (500). Results are accurate.
Figure 2 here CORRECTED
F igure 2.- Chrome GA (first stage optimization) combined chart results for best fit along generations (500). It is shown best fit, average distance between individuals and stopping criteria. Results show be accurate.
Figure 3 here CORRECTED
Figure 3.- Chrome GA (second stage optimization) combined chart results for best fit along generations (500) and stopping criteria. Results show be accurate.

Reviewer 2 Report
The author Francisco Casesnoves has submitted a manuscript entitled "Mathematical Genetic Algorithms to 3D Interior and Inverse Least Squares comparative optimization for standard BCS-parameters Modelling in selected Superconductors and High-Temperature Superconductors" to journal Standards of MDPI.
The introduction provides sufficient background and includes all relevant references. The research design is appropriate. The methods are adequately described. The results are clearly presented. Discussion of data and conclusions are adequately supported by the results.
I have some comments:
1) Line 286: "where MO is the molecular eight" should be "where MO is the molecular weight". Am I right?
2) Figure 1 is a bit difficult to be understood. The quality of the figure is low. Moreover, font size is very small, I would suggest to increase the font size. Finally, I would suggest to put x-axis in a log scale.
Same for Figure 2 and 3.
3) In general, considering the template of the manuscript I think that the figure does not fit very well the template (the width is a bit too large). I would suggest to make the figures a bit more square.
4) In many cases, there are 13 digits as decimals. Does the author think that all these decimals are necessary?
In other cases, the author reports values with only 4 digits. This could make sense for all the values.
Author Response
NOTES TO REVIEWER 2
The corrrections and images that you requested (for example, log scale in Genetic Algorithms images ) are done, see the pdf report,
we are working in all the Reviewer 1 corrections and implementing all at the new paper version,
We will submit also a review report for academic Editor with all of his corrections ,
Dr F Casesnoves PhD

Round 2
Reviewer 1 Report
Much improved this time, thanks!
Maybe want to do somthing about the blank in p.4.